# Regulation of heme utilization and homeostasis in *Candida albicans*

**Natalie Andrawes[1], Ziva Weissman[1], Mariel Pinsky[1], Shilat Moshe[1], Judith Berman[2], Daniel Kornitzer[1]***

**1** Department of Molecular Microbiology, B. Rappaport Faculty of Medicine, Technion–I.I.T., Haifa, Israel,
**2** School of Molecular Microbiology and Biotechnology, Faculty of Life Sciences, Tel Aviv University, Tel Aviv, Israel

* danielk@technion.ac.il

**Data Availability Statement:** All relevant data are within the manuscript and its Supporting Information files. The full RNA-SEQ dataset is available at NCBI's GEO database under accession number GSE202577.

## Abstract

Heme (iron-protoporphyrin IX) is an essential but potentially toxic cellular cofactor. While most organisms are heme prototrophs, many microorganisms can utilize environmental heme as iron source. The pathogenic yeast *Candida albicans* can utilize host heme in the iron-poor host environment, using an extracellular cascade of soluble and anchored hemophores, and plasma membrane ferric reductase-like proteins. To gain additional insight into the *C. albicans* heme uptake pathway, we performed an unbiased genetic selection for mutants resistant to the toxic heme analog $Ga^{3+}$-protoporphyrin IX at neutral pH, and a secondary screen for inability to utilize heme as iron source. Among the mutants isolated were the genes of the pH-responsive RIM pathway, and a zinc finger transcription factor related to *S. cerevisiae HAP1*. In the presence of hemin in the medium, *C. albicans HAP1* is induced, the Hap1 protein is stabilized and Hap1-GFP localizes to the nucleus. In the *hap1* mutant, cytoplasmic heme levels are elevated, while influx of extracellular heme is lower. Gene expression analysis indicated that in the presence of extracellular hemin, Hap1 activates the heme oxygenase *HMX1*, which breaks down excess cytoplasmic heme, while at the same time it also activates all the known heme uptake genes. These results indicate that Hap1 is a heme-responsive transcription factor that plays a role both in cytoplasmic heme homeostasis and in utilization of extracellular heme. The induction of heme uptake genes by *C. albicans* Hap1 under iron satiety indicates that preferential utilization of host heme can be a dietary strategy in a heme prototroph.

## Author summary

The yeast *Candida albicans* is a human commensal organism, as well as an important opportunistic systemic pathogen. During tissue invasion, systemic pathogens are confronted with iron scarcity, which they can overcome by scavenging host heme as iron source. It was however not known whether *C. albicans* can sense the presence of host heme independently of iron sensing. Using a forward genetics approach, we identified a transcription factor that regulates both homeostasis of internal heme and uptake of external heme. This transcription factor allows activation of the heme uptake pathway even in

**Funding:** This study was supported by Israel Science Foundation (https://isf.org.il) grant 587/19 to DK. The funders had no role in study design, data collection and analysis, decision to publish, or preparation of the manuscript.

**Competing interests:** The authors have declared that no competing interests exist.

iron-rich medium, suggesting that heme can be a preferred iron source over elemental iron, and that heme prototrophs can scavenge host heme as a source for cellular heme.

## Introduction

Heme (iron protoporphyrin IX) is an essential cofactor that is required by almost all known aerobic organisms [1]. As a cofactor, heme mediates numerous catalytic, binding and regulatory functions in the cell [1,2]. However, the hydrophobicity and redox activity of heme make it potentially cytotoxic due to mis-incorporation into certain proteins, disruption of lipid bilayers, or oxidation of biomolecules [3,4].

Most aerobic organisms have the capacity to synthesize heme *de novo* [5]. On the other hand, some organisms, notably protozoan parasites such as Leishmania [6], and some metazoans such as the worm *C. elegans* [7], are heme auxotrophs and must assimilate exogenous heme sources for all heme-dependent processes. Among prototrophs, heme may also be acquired from exogenous sources, albeit the extent to which this occurs is cell type- and organism-dependent [1,8]. Furthermore, it is unclear to which degree exogenous heme may be used intact, versus being catabolized to iron that is then recycled for new heme synthesis or any other iron-dependent processes [1].

Iron acquisition is a central challenge for pathogenic microorganisms, which must contend with an environment particularly scarce in iron, due to sequestration mechanisms that drastically limit iron bioavailability in the host [9]. In the human host, ~70% of the iron is bound by the oxygen carrier hemoglobin, as part of its heme prosthetic group [10]. Consequently, many pathogenic microorganisms, bacterial as well as fungal, have evolved mechanisms to exploit host heme, particularly from hemoglobin, as sources of iron [11–13].

*Candida albicans* is a commensal organism of the gastrointestinal tract. It is also one of the most prevalent fungal pathogens, causing superficial mucosal membrane infections in immunocompetent and immunocompromised individuals, as well as life-threatening, systemic infection among immunocompromised or debilitated patients. *C. albicans* can utilize exogenous heme as a source for iron, and furthermore, growth of a *C. albicans* mutant defective in heme synthesis was efficiently restored with exogenous heme, indicating that host heme can potentially be used as heme source as well [14,15].

The *C. albicans* heme acquisition system relies on a cascade of extracellular CFEM (Common in Fungal Extracellular Membrane) hemophores, both soluble and cell wall- or cell membrane-bound, and on the ESCRT pathway, endocytosis, and membrane ferric reductase-like proteins [16–19]. The CFEM gene *RBT5* is among the most highly induced genes in experimental animal infection models [20], and Rbt5-specific antibodies are prominent in the serum of patients recovering from candidemia [21], confirming the importance of the CFEM heme-acquisition pathway in infection. The three secreted CFEM proteins involved in hemoglobin-iron acquisition, Csa2, Rbt5 and Pga7, are soluble, anchored to the outer–, and to the inner cell envelope, respectively. The observation that these three proteins can all extract heme from hemoglobin and transfer it from one protein to the next [18,19] is consistent with a heme transfer cascade across the cell envelope, followed by internalization via endocytosis [11,17]. Recently, two ferric reductase-like proteins, Frp1 and Frp2, also were identified as necessary for heme utilization and for sensitivity to toxic heme analogs [22].

*C. albicans* is normally a diploid organism lacking a full sexual cycle, making it refractory to forward genetic analysis. However, a stable haploid strain recently described [23], in conjunction with transposon mutagenesis systems [24–26], have made this organism amenable to

forward genetics. We took advantage of this system to identify new components and regulators of the heme utilization pathway in *C. albicans*. One of the genes identified, which we characterize here, encodes a zinc finger transcription factor that responds to extracellular heme and regulates both cytoplasmic heme homeostasis and extracellular heme uptake.

## Results

### Identification of mutants defective in hemoglobin utilization

Identification of mutants unable to utilize hemoglobin was done in two stages. In the first stage, we enriched a pool of haploid cells mutagenized with the Ac/Ds transposon from maize [24] for $Ga^{3+}$-protoporphyrin IX (GaPPIX)-resistant clones. GaPPIX is a toxic heme analog that inhibits the growth of bacteria and yeasts [27,28]; in *C. albicans*, GaPPIX toxicity depends on a ferric reductase-like protein involved in heme uptake, Frp2 [22]. The mutagenized pool was serially diluted 1:1000 and outgrown five times in YPD medium at pH 7.5, supplemented with 1 mM of the iron chelator ferrozine to elicit expression of heme uptake genes, and containing 150 μM GaPPIX. In a second stage, aliquots from the GaPPIX enrichment pools were spread on YPD plates and then replica-plated on YPD plates supplemented with the iron chelator BPS (bathophenanthroline disulfonic acid) and hemoglobin, a medium that compels the cells to use hemoglobin as iron source. Mutants unable to grow on these plates, i.e. unable to use hemoglobin as the sole iron source, were isolated and characterized for their transposon insertion site. Since we found that the mutant pools after enrichment round 3 all contained insertions in *FRP2* only, we screened for hemoglobin utilization mutants only after the first two rounds of enrichment.

Table 1 shows the identification of the transposon insertions that were associated with reduced growth on hemoglobin. The vast majority (62/73) fell in eight genes that constitute

**Table 1. Transposon insertion sites.**

| Gene | # of occurrences (# of independent insertion sites) |
|---|---|
| RIM8 | 7 (6) |
| RIM9 | 4 (3) |
| RIM13 | 6 (5) |
| RIM20 | 14 (12) |
| RIM21 | 20 (13) |
| orf19.2914/RIM23 | 5 (4) |
| RIM101 | 2 (2) (both in promoter) |
| DFG16 | 4 (3) |
| VPS23 | 1 |
| VPS25 | 1 |
| VPS36 | 2 (2) |
| **HAP1/ZCF20** | **2 (2)** |
| Orf19.5918 | 1 |
| Orf19.1813/FLC2 | 1 |
| Orf19.703/CPD1 | 1 |
| RTG3 | 1 |
| MRP2 | 1 |
| MSH6 | 1 |
| FRP2 | 1 |

List of genes with transposon insertion displaying reduced hemoglobin utilization.

the RIM pathway, a signal transduction pathway that is activated in neutral or alkaline pH [29] and that was shown to activate, among others, *FRP1* [30,31], which is essential for heme-iron acquisition [22]. Three other genes, *VPS23*, *VPS25*, *VPS36*, encode components of the ESCRT pathway, which was shown to be involved in hemoglobin-iron acquisition [17]. Seven other genes occurred once, including *FRP2*. *FRP2* is the gene most strongly selected for by GaPPIX, however *FRP2* mutants are generally not defective in heme uptake in unbuffered YPD medium [22]. A single insertion, near the 3' end of the coding region, was an exception, for unknown reasons.

One more gene, orf19.4145 / C5_01500C / *ZCF20*, a predicted zinc finger transcription factor, was identified by two transposon insertions, and is characterized below as a heme-regulated activator of genes for heme uptake and homeostasis. The sequence most similar to Zcf20 in the *S. cerevisiae* genome is Hap1 (Heme Activator Protein 1), a heme-regulated transcription factor [32–34]. However, the similarity between the protein sequences is low, and is mainly concentrated in the DNA binding domain of Hap1 (S1 Fig). Furthermore, a search for Hap1-related sequences in *C. albicans* and related species of the CUG-Ser1 clade, showed that the Mrr1 transcription factor and its relatives are equally or more closely related than Zcf20 and its relatives to *S. cerevisiae* Hap1 (S2 Fig). However, closer examination of the sequences revealed that Zcf20 and its relatives contain CP motifs embedded in the consensus K/RCPV/I, which were characterized as heme regulatory motifs (HRM) in Hap1 [35] (S1 and S3 Figs). While Hap1, Zcf20 and all their closest homologs contain multiple repeats of this motif (S3 Fig), the Mrr1-related sequences contain none. Based on this, and based on the analyses shown below, we suggest that *ZCF20* is the likely ortholog of *S. cerevisiae HAP1*, and we therefore propose to rename it *C. albicans HAP1*.

## Hap1 promotes sensitivity to heme analogs and hemoglobin-iron utilization

To confirm the *hap1* phenotypes in a standard diploid *C. albicans* strain, we first took advantage of an existing deletion of *HAP1/ZCF20* within a library of transcription factor mutants [36], and found that the *hap1*⁻/⁻ strains are defective in growth at low hemoglobin concentrations compared to the wild-type (S4 Fig). For further experiments, we constructed a new set of *HAP1* deletion strains and reintegrants in a standard diploid strain background. We tested sensitivity of the mutants to GaPPIX in several ways. First, we replicated the initial screen and grew the wild-type and two independent mutants in shaking cultures with 150 μM GaPPIX. Under these conditions, the *hap1*⁻/⁻ deletion strains grew faster than the wild-type (S5 Fig). We then measured the minimal inhibitory concentration (MIC) of GaPPIX as well as of another toxic heme homolog, $Co^{3+}$-protoporphyrin IX (CoPPIX), in 96 well plate format. MIC for GaPPIX was increased several-fold in the *hap1*⁻/⁻ mutant strains (Fig 1A), and the *hap1*⁻/⁻ mutant appeared to be completely resistant to up to 0.2 mM CoPPIX (Fig 1B). The *hap1*⁻/⁻ mutant was similarly much more resistant to toxic heme analogs in Petri plates (Fig 1C).

We next tested heme-iron utilization of the *hap1*⁻/⁻ mutant vs. wild-type or reintegrant strains. The mutant exhibited reduced growth on plates containing 1 μM hemoglobin as the sole iron source (Fig 2A). In liquid medium, growth of the mutant strains was reduced at low concentrations but restored at higher concentrations of either hemoglobin or hemin as iron sources (Fig 2B and 2C).

## Hap1 drives hemin-responsive gene expression

To identify potential Hap1 target genes in the *C. albicans* genome, we scanned the intergenic regions of the *C. albicans* genome for the proposed consensus binding site of *S. cerevisiae*

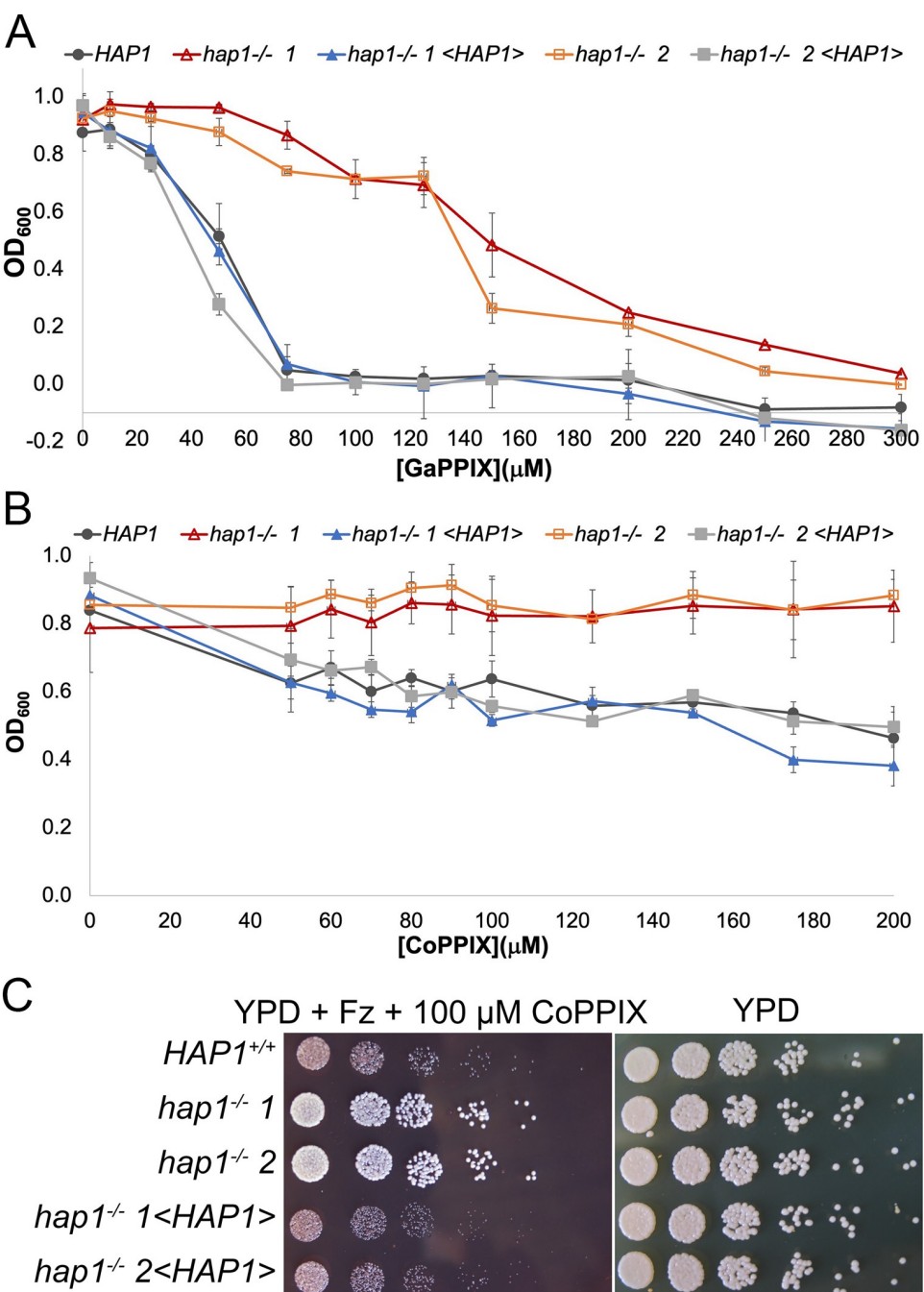

**Fig 1. The *hap1*⁻/⁻ mutant is resistant to the heme analogs GaPPIX and CoPPIX.** (A) GaPPIX MIC (minimal inhibitory concentration) assay of the wild-type (KC1266), two independent *hap1*⁻/⁻ deletion strains (KC1408) and the reintegrated strains (KC1407). The indicated strains were inoculated in triplicate with the indicated GaPPIX concentrations in 96 well plates, in YPD + 1 mM ferrozine to induce the heme-uptake genes, and incubated for 2 days at 30˚C. The graph indicates the average density for each triplicate, and the error bars indicate the standard deviations. (B) CoPPIX MIC assay of the same strains as in A. (C) Drop dilution assay of the same strains, inoculated on YPD medium supplemented with 1 mM ferrozine and 0.1 mM CoPPIX, vs. a YPD control plate. The Petri plates were incubated at 30˚C for two days.

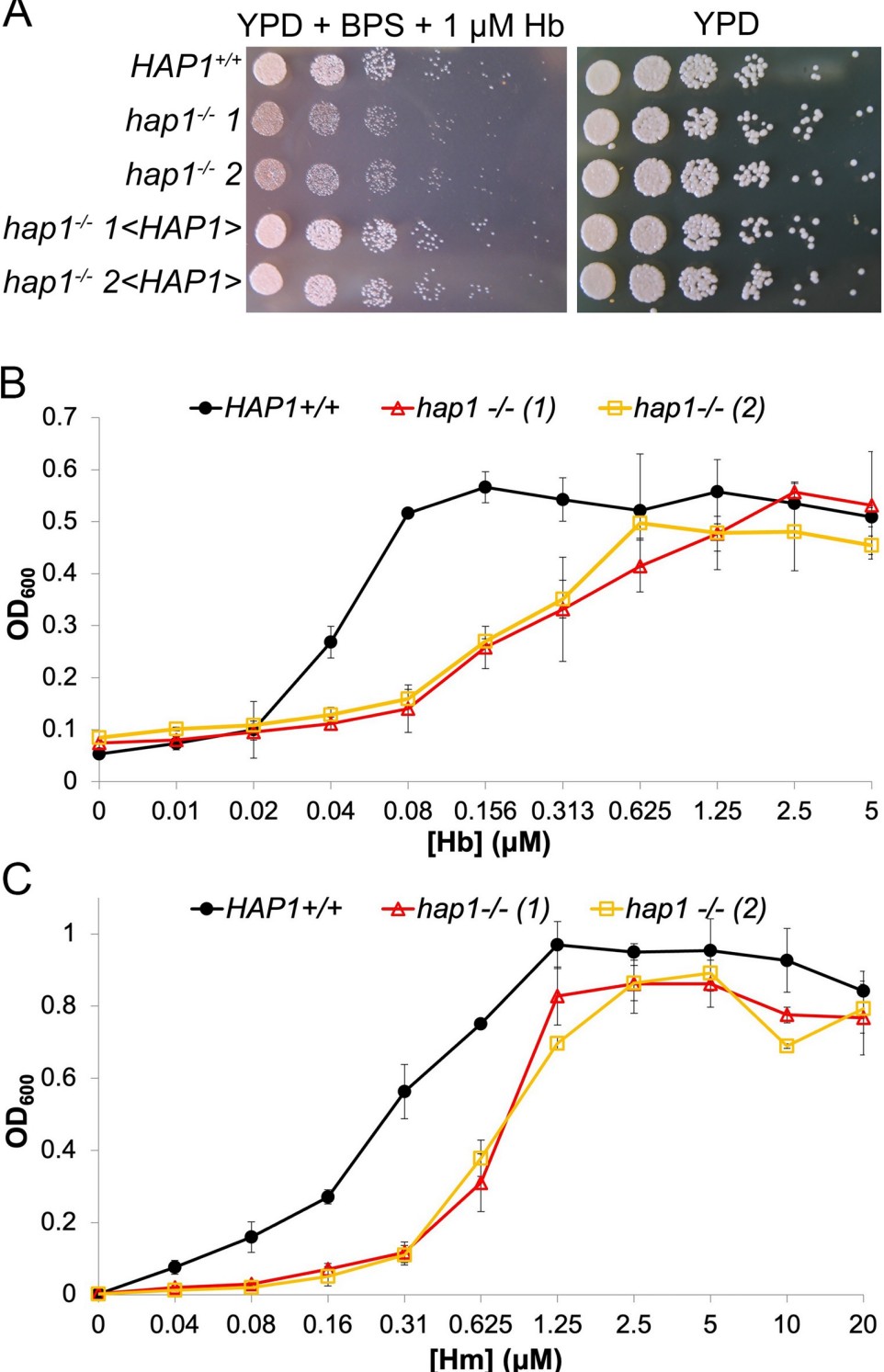

**Fig 2. The *hap1⁻/⁻* mutant is defective in growth on hemoglobin and hemin as iron sources.** (A) Drop dilutions of the wild-type and mutant *hap1* strains on YPD plates supplemented with the iron chelator BPS, which prevents growth of *C. albicans* due to iron limitation, and with 1 μM hemoglobin. The same strains were used as for Fig 1, and the Petri plates were incubated at 30°C for two days. (B,C) The wild-type and mutant strains were inoculated in triplicate with the indicated hemoglobin (B) or hemin (C) concentrations in 96 well plates, in YPD + BPS, and incubated for 2 days at 30°C. The graph indicates the average density for each triplicate, and the error bars indicate the standard deviations.

**Fig 3. *HAP1* affects expression of *HMX1* and *FRP2* and is itself induced by hemin.** A wild-type (SN148) and two mutant strains (KC1406) exposed to 50 μM hemin in the medium, were tested for expression of the indicated genes by RT-PCR at the indicated times after exposure. Expression of each sample was normalized to actin. The values are averages of three independent cultures, with error bars indicating standard deviations.

Hap1 (CGGNNNTANCGG [35,37]), using the Patmatch function of the Candida Genome Database site. We found 13 sites, including one near *FRP2* (pos. -324) and one near *HAP1* itself (pos. -878). The heme oxygenase *HMX1*, an additional candidate Hap1 target gene based on its function, has a near-consensus site at -206. Based on the presence of the *S. cerevisiae* Hap1 consensus site, we surmised that these genes might be activated by Hap1 in the presence of hemin.

To test Hap1-dependent expression of these genes in the presence of hemin, a wild-type strain and two *hap1*[-/-] mutant strains were exposed to 50 μM hemin and samples were taken for qRT-PCR at 0, 1 h and 2 h after hemin addition. The results shown in Fig 3 indicate that in the wild-type strain, all three genes (*FRP2*, *HMX1* and *HAP1*) are induced in the presence of hemin. Furthermore, *HMX1* expression is largely dependent upon Hap1 even in the absence of hemin. In contrast, *FRP2* is expressed in the absence of Hap1, but at lower levels than in the wild-type strain. As expected, no *HAP1* expression is detected in the *hap1*[-/-] mutant. However, *HAP1* is itself induced in the presence of hemin in the medium.

## Hap1 regulates heme homeostasis and heme influx into the cell

The effect of Hap1 on expression of *FRP2* suggests that it is involved in heme uptake, whereas its effect on the heme oxygenase *HMX1* suggests that it is involved in cytoplasmic heme homeostasis. To test the influence of Hap1 on cytoplasmic heme levels in the absence and presence of added heme in the medium, we used a recently developed cytoplasmic fluorescent heme sensor (HS1). This sensor is comprised of a green fluorescent protein (GFP) domain into which a heme-binding cytochrome domain is embedded, as well as a separate red fluorescent protein (mKATE2) domain. Binding of heme to the cytochrome domain causes quenching of the GFP fluorescence signal but not of the mKATE2 signal; hence, the GFP/mKATE2 fluorescence ratio gives an estimate of cytoplasmic "free" heme levels (or more accurately, "labile" or "exchangeable" heme levels [38,39]), with lower ratios corresponding to higher heme levels [14,40]. To extend the dynamic range of detection, a second sensor, with lower affinity due to a mutation (M7A) in the cytochrome domain, is also used.

We first compared wild-type *HAP1* cells to two independent *hap1*[-/-] mutant strains, each transformed with the two heme sensor plasmids. As expected, the high-affinity HS1 sensor in unperturbed cells gives a lower fluorescence ratio, corresponding to a higher heme occupancy, than the medium-affinity M7A sensor (Fig 4A). Furthermore, in the absence of added hemin to the medium, the *hap1*[-/-] strains exhibited 25% lower fluorescence ratios with the high-affinity HS1 sensor and 15% lower fluorescence ratios with the medium-affinity M7A sensor, small but statistically significant differences that indicate the presence of higher heme concentrations in the cytoplasm of the *hap1*[-/-] cells (Fig 4A). When heme was added to the medium at increasing concentrations, the fluorescence ratios decreased in all cells, indicating influx of heme into the cytoplasm. At all hemin concentrations, the *hap1*[-/-] cells exhibited lower fluorescence

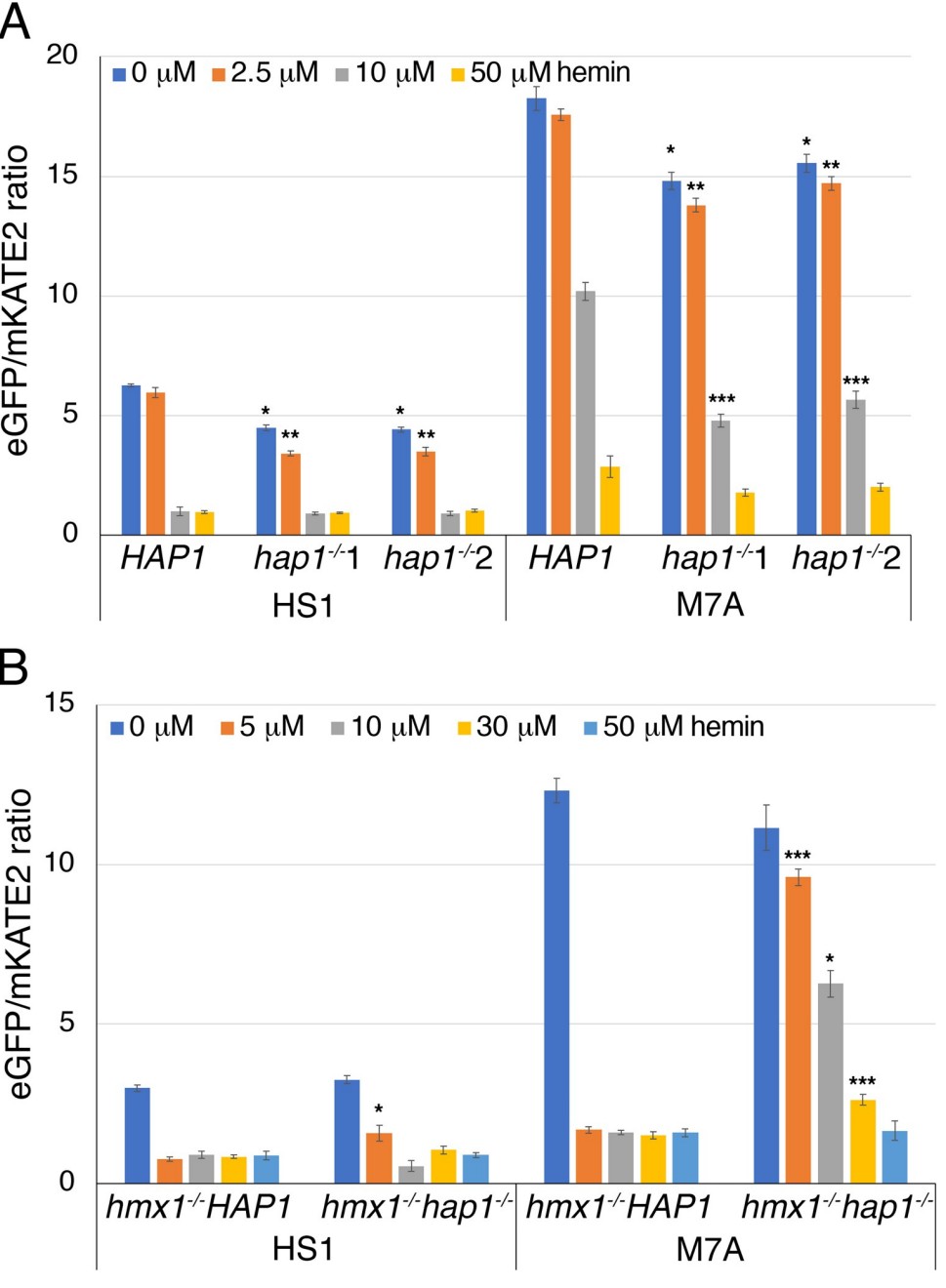

**Fig 4. *HAP1* affects cytoplasmic heme homeostasis and uptake of external heme.** (A) Cells of the indicated phenotype (*HAP1* = SN148, *hap1*$^{-/-}$ = KC1406) were transformed with the HS1 (high-affinity heme sensor)—or M7A (medium-affinity heme sensor) -expressing plasmids and grown in YPD + 1 mM ferrozine and the indicated hemin concentrations for 5 hours. Each data point represents the average of three independent cultures, and the error bars represent the standard deviation. The statistical significance of each mutant data point compared to the corresponding wild-type data point is indicated as * = p<10$^{-4}$, ** = p<10$^{-6}$, *** = p<10$^{-8}$. (B) Cells of the indicated phenotype (*hmx1*$^{-/-}$ *HAP1* = KC1251, *hmx1*$^{-/-}$ *hap1*$^{-/-}$ = KC1426) were transformed with the HS1 or M7A sensor plasmids and grown and assayed as in A.

ratios, i.e. higher cytoplasmic heme concentrations, compared to the wild-type: in the presence of 2.5 μM hemin in the medium, the high and medium affinity sensors gave 40% and 20% lower signals, respectively, in the *hap1*$^{-/-}$ mutant strains vs. the wild-type strain. In 10 μM and

50 μM hemin, the high-affinity HS1 sensor was saturated in all strains, whereas the medium-affinity M7A sensor gave 50% and 35% lower signals in the *hap1*$^{-/-}$ strains (Fig 4A).

We hypothesized that the higher heme levels detected in the *hap1*$^{-/-}$ strains might be because Hap1 affects the expression of *HMX1*, which encodes a cytoplasmic heme-degrading enzyme (Fig 3). To test this hypothesis, we deleted *HMX1* in both the *HAP1* wild-type and mutant strains, and compared heme sensor activity in the absence and presence of hemin in the medium. In the absence of hemin, the HS1 sensor gave identical signals in the *HAP1* wild-type and mutant strains, suggesting that the effect of *HAP1* on "baseline" cytoplasmic heme levels is due to its effect on *HMX1* expression (Fig 4B). We also noted that the HS1 signals were lower in the *hmx1*$^{-/-}$ vs. the *HMX1* strains (compare panels A and B in Fig 4), corresponding to higher baseline cytoplasmic heme levels in the *hmx1*$^{-/-}$ cells, as had been described before [14].

We next tested the effect of addition of 5–50 μM of hemin to *HAP1* and *hap1*$^{-/-}$ strains in the *hmx1*$^{-/-}$ background. The high-affinity HS1 sensor became saturated in both the *HAP1* and *hap1*$^{-/-}$ strains even at the lowest hemin concentration (Fig 4B). The medium-affinity M7A sensor also became saturated at all hemin concentrations in the *HAP1* strain, indicating that the combination of heme influx and lack of degradation by Hmx1 lead to high cytoplasmic heme concentrations. However the *hap1*$^{-/-}$ strain gave a different picture: there, decreasing signals were detected with increasing external hemin, but except for the highest hemin concentration (50 μM), the signals were significantly higher in the *hap1*$^{-/-}$ strain than in the *HAP1* strain, indicating reduced influx of heme into the cell (Fig 4B).

Taken together, the data indicate that Hap1 affects cytoplasmic heme homeostasis *via* its effect on expression of the heme oxygenase *HMX1*. Once *HMX1* is removed, a second effect of Hap1 is revealed, on heme influx: in the *hap1*$^{-/-}$ mutant, less heme appears in the cytoplasm of cells exposed to hemin in the medium than in the *HAP1* wild-type.

## Hap1 is nuclear localized and stabilized in the presence of hemin

To investigate the regulation of Hap1 by heme, we fluorescently tagged the protein at its C-terminus by integration of GFP into one of the two *HAP1* alleles. Cells expressing Hap1-GFP were grown in iron-limiting medium (YPD + 1 mM ferrozine), which induces *HAP1* /orf19.4145 [41], as well as in the presence of hemin, which was shown here to induce *HAP1* (Fig 3), and compared to growth in regular YPD medium. Both iron limitation and the presence of hemin in the medium did induce a fluorescence signal that appeared to be nuclear, whereas the signal was barely visible in regular medium (Fig 5A). Comparison with a nuclear stain confirmed that the GFP signal induced by hemin is indeed concentrated in the nucleus (Fig 5B). Since *HAP1* affects sensitivity to the toxic heme analogs GaPPIX and CoPPIX, we also tested induction of the nuclear Hap1-GFP signal with these and other heme analogs. As shown in Fig 5C, at an identical concentration of 50 μM, hemin gave the strongest induction, but other heme analogs also induced Hap1-GFP, with $Zn^{2+}$-protoporphyrin IX (ZnPPIX) and CoPPIX giving stronger signals than GaPPIX, while $Mn^{3+}$-protoporphyrin IX (MnPPIX) did not appear to induce Hap1-GFP at this concentration.

The induction of the Hap1-GFP signal could be due to induction of *HAP1* transcription, but we surmised that the protein could also be directly affected by the presence of heme. We therefore measured the stability of the protein by placing an epitope-tagged version of *HAP1* under the glucose-repressible *MAL2* promoter and measuring protein levels after addition of glucose. As shown in Fig 5D, while in regular medium, after an initial lag, Hap1 decayed with a half-life of 20–30 min, in the presence of hemin in the medium, it was completely stabilized. Thus, we conclude that stabilization of the Hap1 protein contributes to the increase in Hap1 levels in the presence of hemin in the medium.

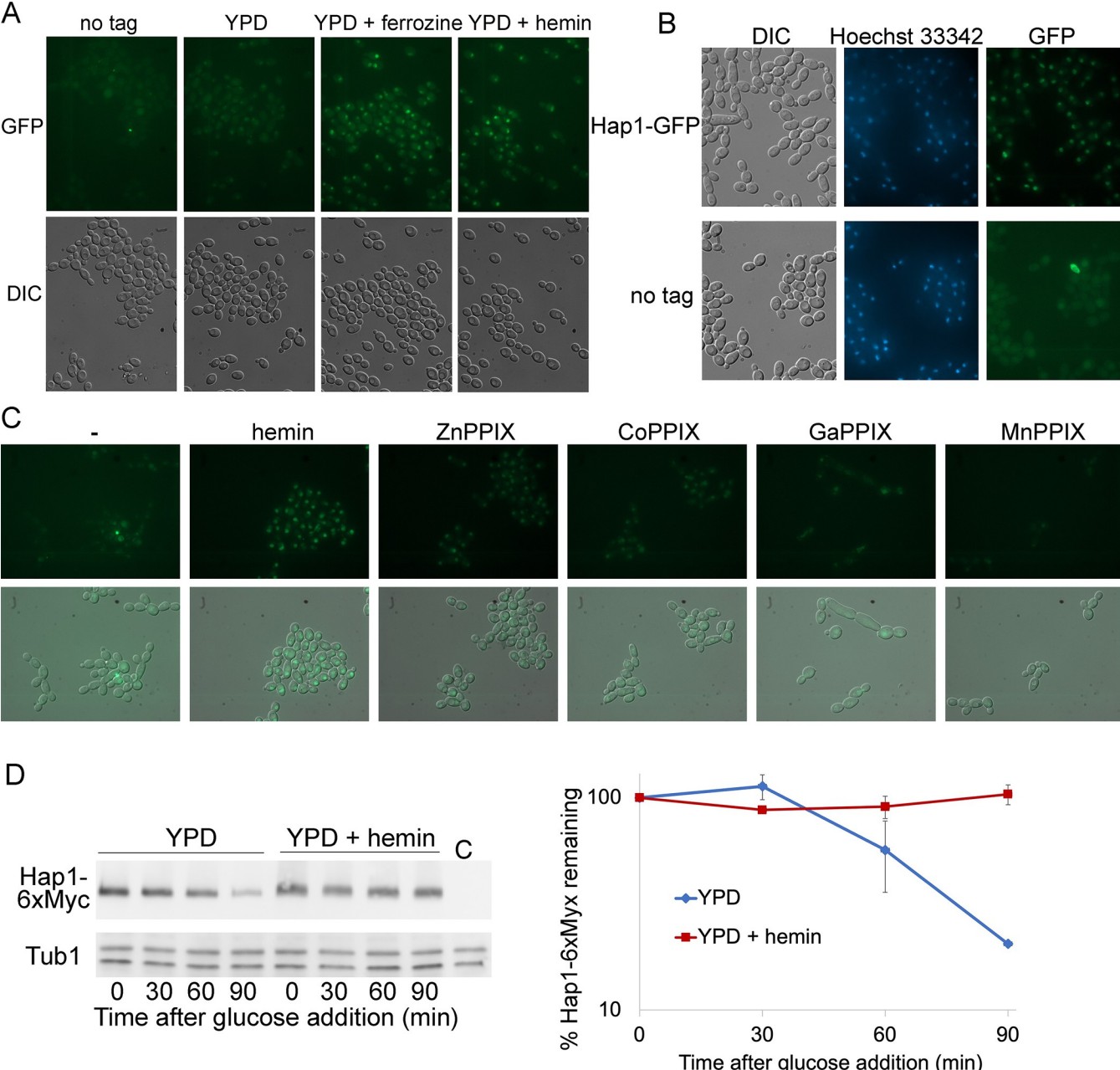

**Fig 5. Regulation of the Hap1 protein.** (A) *HAP1-GFP* is induced by iron starvation and by hemin. Cells expressing a *HAP1-GFP* allele from its native promoter (KC1368) were grown in regular YPD medium, or in YPD supplemented with 1 mM ferrozine or with 50 μM hemin for 4 hours, then visualized by epifluorescence microscopy. "No tag" indicates the untagged original CAI4 cells. (B) Hap1-GFP is located in the nucleus. The strain expressing Hap1-GFP under its native promoter, and a control untagged strain, were grown 4 h in YPD medium containing 1mM ferrozine and 10 μM hemin, then briefly stained with the DNA stain Hoechst 33342 and visualized by epifluorescence microscopy. (C) *HAP1-GFP-* expressing cells were grown 3 h in YPD supplemented with the indicated compounds at a uniform concentration of 50 μM, and visualized by epifluorescence microscopy. Top panels are GFP immunofluorescence micrographs, and bottom panels are immunofluorescence overlaid over DIC micrographs. (D) Hap1 is stabilized in the presence of hemin. Wild type cells carrying MAL2p-Hap1-6xMyc (KC1399), or a control strain (C = SN148) were grown overnight in YEP+ 2% raffinose at 30˚C. In the morning, cells were diluted to $OD_{600} = 0.5$ in YEP+2% maltose, with or without 20 μM hemin, and incubated for 4 hours at 30˚C. Next, 2% glucose was added, and the Hap1-6xMyc protein levels were followed by Western blotting. Time 0' is immediately after glucose addition. Hap1-6xMyc was detected with the anti-Myc antibody 9E10, and the same membrane was reacted with an anti-tubulin antibody. Protein signals were visualized by chemoluminescence and quantitated. The left panel shows the membrane, and the right panel shows the ratio of Myc signal to tubulin signal, normalized to the signal at time 0' for each culture (= 100%). The graph shows the average of two separate experiments. The error bars indicate standard deviations.

## Role of Hap1 in global gene expression in the presence of hemin

To identify the genes activated by Hap1 in the presence of hemin in the medium, *hap1*⁻/⁻ cells transformed either with a vector plasmid or with a plasmid harboring the wild-type *HAP1* gene were exposed to 50 μM hemin in the medium for 30 min, and their global transcriptome was analyzed by RNA-SEQ (S1 Table). Expression was compared between *HAP1* and *hap1*⁻/⁻ cells with or without added hemin, and between hemin-supplemented medium and control medium in both types of cells (see S2 and S3 Tables for lists of genes significantly induced or repressed, respectively).

To obtain an overview of the types of genes affected by hemin or by *HAP1*, we performed GO enrichment analysis by biological process on all genes induced or repressed 2x or more in *HAP1* reintegrant vs. *hap1*⁻/⁻ mutant cells with or without added hemin, and in hemin-supplemented vs. regular YPD medium in both types of cells (S4 and S5 Tables). Fig 6A lists the top GO terms of the Biological Process hierarchy in each comparison. Addition of hemin in the *HAP1* strain caused an increase, among induced genes, in "oxidant detoxification", consistent with the notion that free heme can cause oxidative damage to membranes and proteins [3], as well as "response to host immune response", whereas hemin addition caused an increase in "carbohydrate transport" both in the presence and absence of Hap1. Addition of hemin also caused repression of ribosome biogenesis., both in the presence and absence of Hap1.

When comparing *HAP1* and *hap1*⁻/⁻ cells, "cellular iron homeostasis" stands out among induced genes, probably reflecting the heme-uptake genes induced by Hap1, as detailed below. In addition, in *HAP1* vs. *hap1*⁻/⁻ cells in regular medium, rRNA biogenesis also appears among the "induced" genes—i.e., repressed in *hap1*⁻/⁻—, whereas categories representing ATP generation by glycolysis, carbohydrate transport and responses to toxic chemicals appear among the repressed genes—i.e., induced in *hap1*⁻/⁻. In the presence of added hemin in the medium, in contrast, most of these GO categories are not overrepresented anymore in *HAP1* vs. *hap1*⁻/⁻ cells.

The overlap between the four subsets of genes induced at least 2x in all four pairwise comparisons was plotted in Fig 6B. Only seven genes were induced by hemin and were also dependent on Hap1 in all media, including the four genes (CFEM hemophores *RBT5* and *PGA7*, and *FRP1*, *FRP2*) known to be essential for heme acquisition. Three genes that were induced by hemin but that were dependent on Hap1 only in the presence of hemin, include another CFEM hemophore, CSA1. A much larger group of genes (91, or 38.6%) was induced by hemin both in the wild-type and *hap1*⁻/⁻ strains. GO analysis of this group of genes (Fig 6B; S4 Table, +He vs YPD, *HAP1* ∩ *hap1*⁻) revealed an enrichment in genes involved in the response to reactive oxygen species, and in carbohydrate transport, as seen also in the GO analysis described in panel A. In summary, hemin induces a response to reactive oxygen species and suppresses ribosome biosynthesis independently of *HAP1*, whereas *HAP1* induces mainly genes of metal homeostasis, specifically of uptake and homeostasis of heme.

To obtain a more detailed view of genes dependent on Hap1, in Table 2 we have listed the genes most expressed (3x or more) in hemin-supplemented medium in the presence vs. the absence of *HAP1*. Strikingly, the top of the list is populated by the CFEM hemophores *RBT5*, *PGA7* and *CSA1*, by *FRP1* and *FRP2*, which are ferric reductase-like proteins required for heme uptake, and by the heme oxygenase *HMX1*. In Table 3, we have compared expression of these six genes under all four conditions (in *HAP1* and *hap1*⁻/⁻, with or without added hemin). What emerges from this comparison is that Hap1 is responsible not only for expression of these genes when hemin is added to the medium, but also for baseline expression in the absence of added hemin in the medium (compare *HAP1* YPD to *hap1*⁻/⁻ YPD). Furthermore, in the absence of *HAP1*, exposure to hemin does not appear to induce these genes.

## A

**Addition of hemin** | induced | repressed | **HAP1 vs hap1⁻/⁻**

### HAP1

| | |
|---|---|
| cellular oxidant detoxification | 16.68 |
| response to host immune response | 10.42 |
| carbohydrate transport | 3.72 |
| ribosomal large subunit export from nucleus | 11.04 |
| ribosomal large subunit assembly | 9.58 |
| endonucleolytic cleavage in ITS1 to separate SSU-rRNA from 5.8S rRNA | 5.31 |
| endonucleolytic cleavage to generate mature 5'-end of SSU-rRNA | 5.15 |

### hap1⁻/⁻

| | |
|---|---|
| carbohydrate transport | 3.4 |
| nucleobase catabolic process | 32.96 |
| assembly of large subunit precursor of preribosome | 28.84 |
| ribosomal large subunit assembly | 9.71 |
| maturation of 5.8S rRNA from tricistronic rRNA transcript | 5.42 |
| endonucleolytic cleavage to generate mature 5'-end of SSU-rRNA | 3.54 |

### YPD

| | |
|---|---|
| cellular iron ion homeostasis | 10.66 |
| endonucleolytic cleavage to generate mature 5'-end of SSU-rRNA | 3.51 |
| energy reserve metabolic process | 12.83 |
| glycolytic process | 11.58 |
| response to toxic substance | 4.77 |
| small molecule catabolic process | 3.23 |
| cellular response to chemical stress | 3.01 |
| carbohydrate transport | 2.56 |
| proteolysis | 1.82 |

### YPD + hemin

| | |
|---|---|
| cellular iron ion homeostasis | 16.85 |
| response to toxic substance | 10.77 |

## B

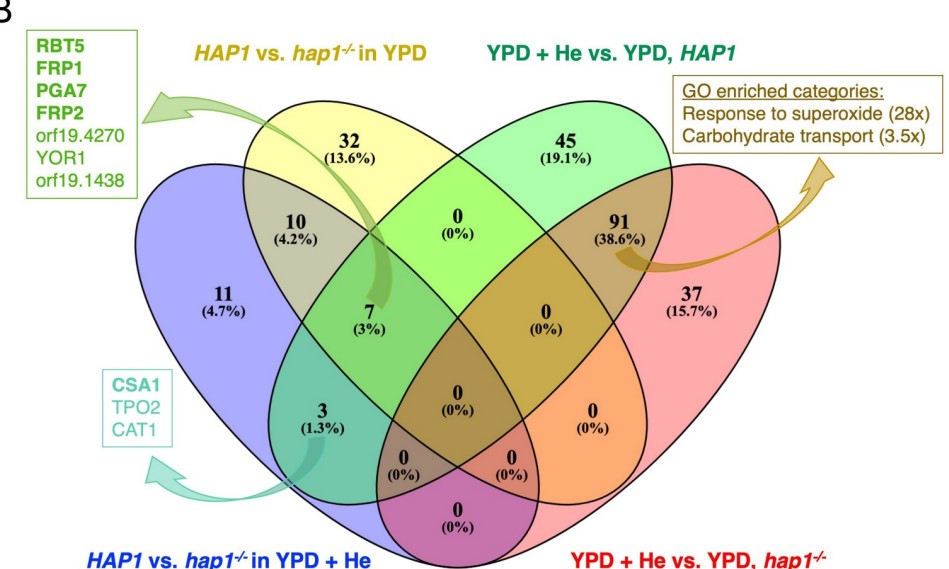

**Fig 6. Analysis of genes regulated by hemin vs. genes regulated by Hap1.** (A) GO analysis of genes induced or repressed at least 2x in all comparisons showed. Only the top term in the GO hierarchy is shown. Full lists are available in S4 and S5 Tables. (B) Venn diagram of overlap of genes induced at least twice. The four groups are: genes induced at least 2-fold in *HAP1* vs. *hap1⁻/⁻* in the presence of hemin (violet) or in regular medium (yellow) and the genes induced at least 2-fold by hemin in the *HAP1* strain (green) or in the *hap1⁻/⁻* strain (pink orange). The boxes on the left give the names of the seven genes induced by hemin and by *HAP1* with or without hemin (dark green), and by hemin and by *HAP1* in the presence of hemin only (turquoise). CFEM hemophores and ferric reductase-like genes *FRP1*, *FRP2* are in bold. The box on the right lists the enriched GO categories of genes overexpressed in the presence of hemin in both *HAP1* and *hap1⁻/⁻* cells. The full lists of the genes in each of the populated categories are shown in supplementary S2 Table.

**Table 2. Genes most dependent on Hap1.**

| orf19. | Gene | Fold change, *HAP1* vs. *hap1*[-/-] | CGGNNNTANCGG | CGGNNNTANCGG (1 mismatch allowed) |
|--------|------|------|------|------|
| orf19.5636 | RBT5 | 357 | | + |
| orf19.5634 | FRP1 | 178 | | + |
| orf19.5635 | PGA7 | 35 | | + |
| orf19.6073 | HMX1 | 14 | | + |
| orf19.7112 | FRP2 | 12 | + | ++ |
| orf19.2449 | | 6.7 | | |
| orf19.4270 | MNN13 | 6.1 | | + |
| orf19.7114 | CSA1 | 5.7 | + | ++ |
| orf19.1783 | YOR1 | 4.6 | | + |
| orf19.5677 | DUR4 | 3.9 | | |
| orf19.1438 | | 3.7 | | + |
| orf19.8278 | | 3.6 | | + |
| orf19.7148 | TPO2 | 3.6 | + | ++ |
| orf19.4377 | KRE1 | 3.5 | | |
| orf19.6229 | CAT1 | 3 | | |

Listed are the genes that are expressed 3-fold or more in *HAP1* reintegrant (KC1407) vs. *hap1*[-/-] cells (KC1408) exposed for 30 min to 50 μM hemin. The two last columns indicate the presence of a perfect Hap1 consensus sequence in the promoter (before-last column) or a consensus sequence with 1 mismatch allowed (last column).

We next asked whether genes induced or repressed in the *HAP1* vs. the *hap1*[-/-] cells were enriched for the putative Hap1 consensus binding site CGGNNNTANCGG [37]. As mentioned above, in the haploid genome, this motif was found at 13 sites in intergenic regions, adjacent to 26 genes (0.4% of the total; S6 Table), including *HAP1* itself. Out of 15 genes most induced by Hap1 (3x and more), the motif is found adjacent to 3 genes (20%) ($p < 0.00003$) (Table 2). In contrast, the motif is not found adjacent to any of the 55 genes repressed 2x or more in *HAP1* vs. *hap1*[-/-] cells. When a single mismatch is allowed, the motif is found adjacent to 1678 genes (28%) and in 11 of the 15 most-expressed genes (73%) ($p < 0.0004$) (Table 2). We conclude that the binding motif of *C. albicans* Hap1 likely resembles that of *S. cerevisiae* Hap1, and that some of the most-highly induced genes in *HAP1* vs. *hap1*[-/-] cells (Table 2) are likely to be direct targets of *C. albicans* Hap1.

### Role of Hap1 in iron starvation-induced gene expression

The CFEM hemophore genes, as well as *FRP1* and *FRP2*, were previously found to be among the most strongly induced genes under conditions of iron limitation [41,42]. Here however,

**Table 3. Expression of six Hap1-dependent genes involved in heme uptake and homeostasis.**

| | *HAP1* YPD | *HAP1* +He | *hap1*[-/-] YPD | *hap1*[-/-] +He |
|------|------|------|------|------|
| RBT5 | 1 ± 0.19 | 7.8 ± 1.7 | 0.041 ± 0.001 | 0.022 ± 0.001 |
| PGA7 | 1 ± 0.16 | 4.3 ± 1.0 | 0.16 ± 0.01 | 0.12 ± 0.01 |
| CSA1 | 1 ± 0.22 | 3.7 ± 0.8 | 0.86 ± 0.06 | 0.64 ± 0.07 |
| FRP1 | 1 ± 0.25 | 3.8 ± 0.8 | 0.067 ± 0.06 | 0.021 ± 0.003 |
| FRP2 | 1 ± 0.28 | 3.0 ± 0.4 | 0.14 ± 0.01 | 0.24 ± 0.01 |
| HMX1 | 1 ± 0.20 | 1.5 ± 0.1 | 0.010 ± 0.04 | 0.10 ± 0.01 |

For each gene, expression (normalized counts of the RNA-SEQ readings) was normalized to expression in the wild-type strain in YPD. Each result is the average of three biological replicates, with standard deviations indicated.

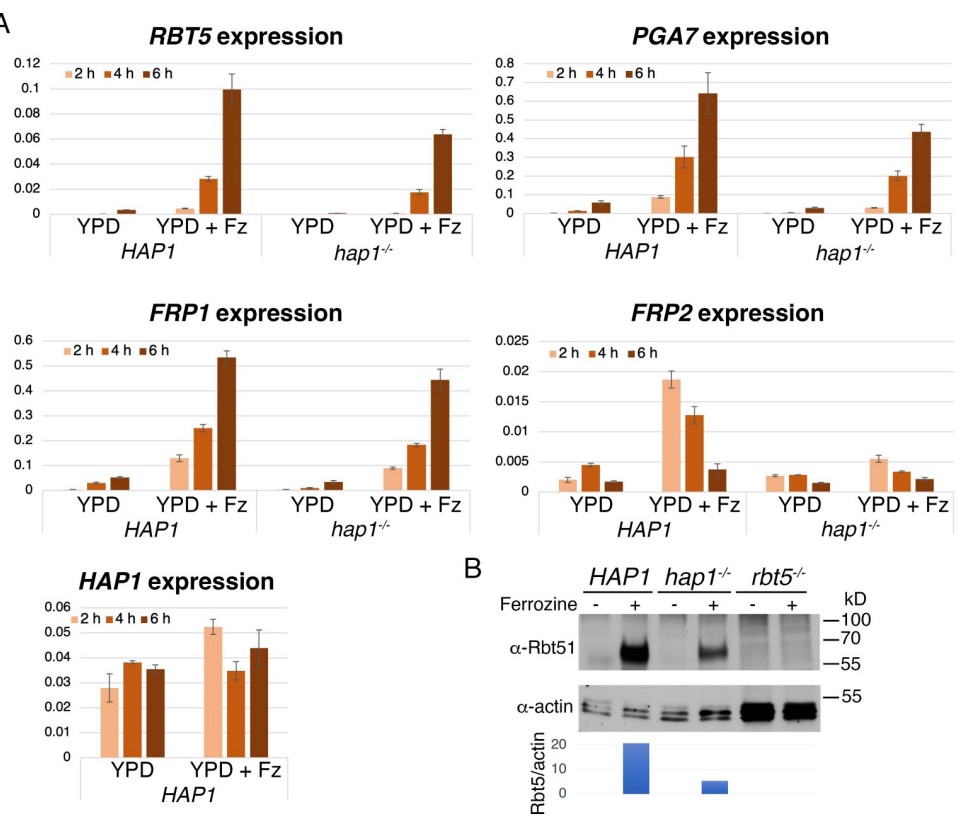

**Fig 7. Role of Hap1 in iron starvation-induced expression of heme uptake genes.** (A) *hap1*[-/-] mutant (KC1408) and reintegrant (KC1407) log phase cultures were kept in YPD or shifted to YPD + 1 mM ferrozine (YPD + Fz) and aliquots were taken after 2, 4 and 6 h. Expression of the indicated genes was quantitated by RT-PCR and normalized to actin. The values are averages of three independent cultures, with error bars indicating standard deviations. (B) *hap1*[-/-] mutant and reintegrant strains, as well as an *rbt5*[-/-] strain (KC108), were grown as in A, aliquots were taken after 4 hours and the Rbt5 protein was detected with a polyclonal antibody (top panel). The actin signal was used as internal control (middle panel), and the bottom panel indicates the ratio of Rbt5 to actin signal.

we have shown that their expression depends upon the heme-regulated Hap1 transcription factor. Thus, to clarify the role of Hap1 in expression of these genes under iron limitation, we measured expression of several of the Hap1 target genes, *RBT5*, *PGA7*, *FRP1* and *FRP2*, as well as *HAP1* itself, under conditions of iron limitation. To impose iron limitation, cells were transferred to medium containing the iron chelator ferrozine, which was previously shown to strongly induce *RBT5* [16].

As shown in Fig 7A, all genes tested were strongly induced by the shift to ferrozine medium, except *HAP1* itself, which showed only a limited induction at the earliest time-point. Kinetics of induction differed: *FRP2*, in particular, showed its greatest induction at early time points, compared to *RBT5* and *PGA7*, which were gradually induced between the 2 h and the 6 h time point. Induction was also seen in the *hap1*[-/-] background, however in all genes tested, the extent of induction was reduced in the absence of Hap1, with the strongest effect detected with *FRP2*. In parallel we also followed the induction of the Rbt5 protein by Western blotting, and found that the amount of Rbt5 detectable upon ferrozine induction was reduced in the *hap1*[-/-] strain compared to the *HAP1* strain, confirming a reduced expression of *RBT5* in the *hap1*[-/-] strain at the protein level as well (Fig 7B). We conclude that while induction of the heme uptake genes by iron starvation does not depend on Hap1, the amplitude of induction was reduced, to different extents in different genes, in its absence.

## Discussion

Organisms respond to their internal and external environment to maintain homeostasis and acquire nutrients. For microorganisms in the animal host, an important environmental factor is iron scarcity, which can be remediated by the acquisition of host heme. Heme, in addition to being itself an essential cellular factor, is also potentially toxic, and its cytoplasmic levels must therefore be closely controlled. We have shown here that *C. albicans* uses a single heme-responsive transcription factor, Hap1, to both maintain cytoplasmic heme homeostasis and activate the utilization of external hemin as heme and iron sources.

*S. cerevisiae* Hap1 is bound and activated by heme [34]. However, rather than being involved in heme uptake like *C. albicans* Hap1, *S. cerevisiae* Hap1 plays a key role in the regulation of gene expression by oxygen [43,44]. Since heme is central to oxidative metabolism and since heme synthesis requires oxygen, the presence of heme is thought to serve as a barometer for the presence of oxygen in the cell [34,45]. Thus, *C. albicans* Hap1 has diverged in function from *S. cerevisiae* Hap1 by acquiring a distinct set of target genes. The two proteins nonetheless appear to have maintained a similar consensus binding site, based on its occurrence in the promoters of the most Hap1-dependent genes (Table 2), however direct binding of Hap1 to these sites remains to be shown experimentally. In addition, while it is not known whether, like *S. cerevisiae* Hap1, *C. albicans* Hap1 binds heme directly, two sequences having the consensus Hap1 heme regulatory motif K/R C P V/I [46] are found in the *C. albicans* Hap1 sequence at positions 288 and 888 (S1 and S3 Figs). Thus, one likely explanation for the stabilization of *C. albicans* Hap1 that we detected when hemin is added to the medium (Fig 5D) is that it is a consequence of direct binding of heme.

On the assumption that binding of heme stabilizes and activates *C. albicans* Hap1, and given that the *HAP1* promoter contains a Hap1 consensus binding site, activation of Hap1 by heme would, in a positive feedback loop, activate its own expression as well as expression of its other target genes. Under normal conditions, when heme is endogenously synthesized, excess heme in the cytoplasm would cause Hap1 to activate the heme oxygenase *HMX1*, which would cause degradation of heme and restoration of homeostasis. Upon exposure of the cell to external heme, an initial influx of heme into the cytoplasm would activate the heme uptake cascade, leading to more heme influx. Our proposed model for the regulation of heme uptake and homeostasis by Hap1 thus consists of a series of interlocked feedback loops (Fig 8). We don't suggest that this model completely describes heme uptake and utilization; for example, unchecked activity of the heme uptake and degradation systems in the presence of heme might, once the cellular heme and iron needs are met, cause iron overload. Preventing this would require the activity of iron-responsive factors to shut off heme uptake genes under iron excess.

Global expression analysis gave insights into the effect of heme overload on the cells, as well as into the specific role of Hap1. GO analysis indicated an activation of genes for oxidant detoxification after exposure to a high concentration of heme, supporting the proposed toxic effect of heme as being due, at least in part, to the generation of reactive oxygen species. An additional general effect of heme exposure was suppression of ribosome biogenesis. This is consistent with previous observations in *S. cerevisiae* that metal stresses as well as other stresses cause suppression of ribosome synthesis genes [47,48]. The oxidant detoxification and suppression of ribosome biogenesis responses that we detected here were not dependent upon Hap1, but could be due to activation of general stress response pathways.

Comparison of gene expression with and without *HAP1* identified a subset of genes involved in cellular iron homeostasis, specifically in heme uptake. Additionally and unexpectedly, even in the absence of added hemin to the medium, the *hap1⁻/⁻* strain showed reduced

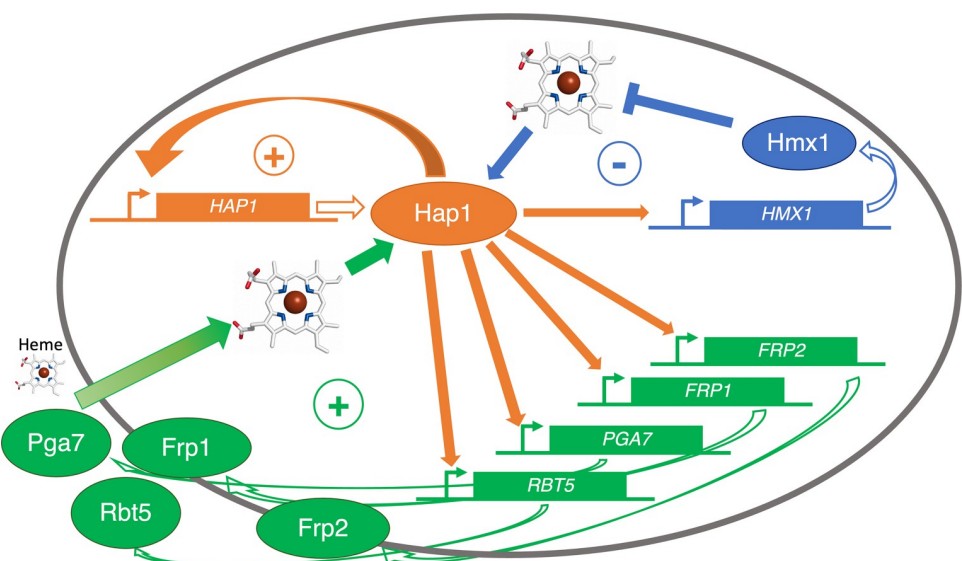

**Fig 8. Schematic model for the regulation of heme uptake and homeostasis by Hap1.** The genes are depicted as rectangles and the gene products as ovals. The heme degradation function of Hmx1 is depicted in blue and the heme uptake function of the CFEM hemophore pathway is depicted in green. The + and - symbols indicate positive feedback loops and negative homeostatic feedback loops, respectively.

rRNA biogenesis, as well as an increase in genes for glycolysis, carbohydrate transport, and response to chemical stress /toxic substance. A possible explanation is that the absence of Hap1 leads to deregulation of heme homeostasis and to higher cellular heme concentrations–as was detected using the HS1 heme sensor (Fig 4A)–which in turn activates the cellular stress response. The increased carbohydrate transport could be related to the shift to glycolysis. It is not clear why the presence of heme should lead to a shift to glycolysis; one possibility is that by shifting away from cellular oxygen consumption and from generation of ROS by the respiratory chain, the cells could mitigate potential heme-catalyzed oxidative damage.

In contrast to the differences between *HAP1* and *hap1*[-/-] cells in regular medium, much less enrichment in GO categories was detected in *HAP1* vs *hap1*[-/-] cells exposed to 50 μM hemin. At such high hemin concentrations, both the wild-type and the mutant cells might experience similar heme overload–compare e.g. cytoplasmic heme in the two types of cells at 50 μM hemin in Fig 4A–, and therefore the general response to high hemin would be unaffected by the absence of *HAP1*.

The CFEM hemophore genes, as well as *FRP1* and *FRP2*, shown here to be regulated by Hap1, had previously been shown to be among the most strongly induced genes under conditions of iron limitation [41,42]. This raised the question whether Hap1 is involved in iron limitation-dependent expression of these genes, in addition to mediating their hemin-induced gene expression. On the one hand, previous data had shown that the iron-responsive transcription factor CBF directly binds the *FRP1* promoter and affects its expression [30]. On the other hand, the screen for GaPPIX-resistant mutants described above was performed under iron-limiting conditions in order to ensure expression of the heme uptake pathway, and nonetheless identified the *HAP1* mutant, suggesting that Hap1 is required even under conditions of gene induction by iron limitation. Furthermore, absence of *HAP1* led to reduced utilization of heme-iron (Figs 2 and S4), even though these experiments are also performed under iron starvation conditions. When we tested the induction of heme uptake genes in iron limited medium in the *hap1*[-/-] mutant, we found that absence of Hap1 somewhat reduced the

amplitude of induction of *RBT5*, *PGA7* and *FRP1*, but strongly reduced the induction of *FRP2*. The effect on *FRP2* explains why *HAP1* was identified in a screen for resistance to GaPPIX, since *FRP2* is essential for full GaPPIX toxicity [22]. However *FRP2* plays a minor role at best in heme-iron utilization in standard medium [22]. Thus, either the partial reduction in expression of *RBT5*, *PGA7* and *FRP1* in the iron-limited *hap1*$^{-/-}$ strain is sufficient to explain the reduced heme-iron utilization, or other, unknown genes regulated by Hap1 participate in heme-iron utilization. Regarding the latter possibility, examination of the genes induced in the presence of Hap1 could reveal new factors involved in fungal heme-iron utilization.

The role of Hap1 identified here, to specifically promote uptake of external heme, underscores the importance of host heme as a micronutrient for animal parasites such as *C. albicans*. Furthermore, although external hemin was shown to be able to serve as source for cellular heme in an artificial heme auxotroph [14], the observation that heme uptake genes are activated via Hap1 even in the presence of sufficient iron in the medium, indicates that external hemin, when available, is actually a preferred heme source for this organism. Given the complexity of heme synthesis [49], this strategy makes metabolic sense, and may turn out to be more widespread than previously thought. External heme uptake must be complemented by a system for transporting the acquired heme from the endocytic system to the different cellular locations where it is required. Our knowledge of intracellular transport mechanisms of externally acquired heme–or, for that matter, of endogenously synthesized heme–is very limited [1,38,50]. Existence of a dedicated uptake system of external heme in *C. albicans* makes it a pertinent organism to probe these questions.

## Methods

### Media and chemicals

Cells were grown in YPD medium (1% yeast extract, 2% bacto-peptone, 2% glucose, tryptophan 150 mg/l) or in Synthetic Complete (SC) medium lacking specific amino acids, as indicated. SC medium contains, per liter, Yeast Nitrogen Base (USBiological) 1.7 g, $(NH_4)_2SO_4$ 5 g, the 20 amino acids, adenine and uridine, 0.1g each except leucine, 0.2 g, glucose 20 g, and 0.2 mM inositol. YPD pH 7.5 was made by adding 0.1 M Tris-Cl pH 7.5 from a 1M stock. RPMI-transferrin was prepared as described previously [16] with the following modifications: RPMI-1640 HEPES (Sigma) was supplemented with 2% glucose, 25 µg/ml uridine and 25 µg/ml adenine, then the medium was desferrated by 5% w/v Chelex100 (Sigma) for 2 hours and 100µg/ml human serum iron chelator apotransferrin (Sigma) was added. Chelex100 beads were removed by filtration and essential metals except iron were restored (40 µg/L $CuSO_4$, 400 µg/L $ZnSO_4$, 400 µg/L $MnCl_2$, 200 µg/L $MoNa_2$ and 200 mg/L $CaCl_2$). The final pH was adjusted to 7.2 with 1N HCl, and only medium prepared the same day was used. Media were supplemented with the ion chelators ferrozine or bathophenanthroline sulfonate (BPS) (Sigma) at 1 mM, hemin as indicated from a 2 mM stock in 50 mM NaOH, or bovine hemoglobin from a 0.5 mM stock in phosphate-buffered saline (Dulbecco's PBS; Biological Industries, Israel). Hemin, $Ga^{3+}$-protoporphyrin IX and $Co^{3+}$-protoporphyrin IX were obtained from Frontier Scientific, and bovine hemoglobin from Sigma (H2500). $Ga^{3+}$-protoporphyrin IX and $Co^{3+}$-protoporphyrin IX were added from a 50 mM stock made fresh in DMSO.

### Strains and plasmids

*C. albicans* strains are listed in Table 4. KC1368 is CAI4 transformed with plasmid KB2753. KC1399 is SN148 transformed with plasmid KB2759. KC1406 was built by sequential deletion of both *HAP1* alleles by homologous recombination with fusion PCR products containing the *NATr* gene and the Cd*LEU2* gene, respectively, each with 500 bp of upstream and downstream

**Table 4. List of *Candida albicans* strains.**

| Name | Genotype | Origin |
|------|----------|--------|
| SN148 | *ura3Δ::imm434/ura3Δ::imm434 his1Δ/his1Δ arg4Δ/arg4Δ leu2Δ/leu2Δ* | [51] |
| CAI4 | *ura3Δ::imm434/ura3Δ::imm434* | [62] |
| KC108 | CAI4 *rbt5::hisG/rbt5::hisG* | [16] |
| KC1251 | SN148 *hmx1Δ::LEU2/hmx1Δ::LEU2* | [14] |
| KC1266 | SN148 *LEU2/leu2Δ ADE2/ade2:: URA3* | This work |
| KC1330 | SN148 *ura3Δ::imm434/URA3 hap1Δ::HIS1/hap1Δ::LEU2* | [36] |
| KC1368 | CAI4 *HAP1/HAP1-GFP URA3* | This work |
| KC1399 | SN148 *ADE2/ade2::MAL2-HAP1-6xMyc URA3* | This work |
| KC1406 | SN148 *hap1Δ::NATʳ/hap1Δ::LEU2* | This work |
| KC1407 | KC1406 *ADE2/ade2::HAP1 URA3* | This work |
| KC1408 | KC1406 *ADE2/ade2::URA3* | This work |
| KC1426 | KC1406 *hmx1Δ::HIS1/hmx1Δ::HIS1* | This work |

*HAP1* sequences (S7 Table, primers 11–14), according to published protocol [51]. KC1407 is KC1406 transformed with plasmid KB2761, and KC1408 is KC1406 transformed with BES116 [52]. KC1426 is KC1406 with both alleles of *HMX1* substituted with the *LEU2* marker, using homologous recombination as above, combined with transient CRISPR as described [14,53].

Plasmid KB2753 contains the 3' end of *HAP1*(+1258 to +3255; primers 15, 16) SpeI–XhoI, cloned into KB2430 digested with the same enzymes, fusing the *HAP1* ORF to GFP. The plasmid is digested with BglII for integration at the chromosomal locus of *HAP1*. KB2430 is the CaGFPgamma and *URA3* XhoI-KpnI fragment from pFA-GFP-URA3 [54] cloned into pBSIISK+. KB2759 was constructed by substituting the PstI-HindIII *GCN4* open reading frame fragment of plasmid KB1575 [55] with the *HAP1* open reading frame (primers 7, 8). KB2761 contains the *HAP1* gene fragment extending (-1890) to (+3636), SpeI and HindIII into BES116 (primers 9, 10).

## Transposon insertion identification

Single colonies were grown overnight in YPD medium, DNA was extracted as described [56], and subjected to the FPNI DNA amplification protocol [57] using Ds-specific primers (S7 Table, primers 1–6) as described [25]. PCR reactions were tested for the presence of bands on an agarose gel and the positive reactions (>90%) were cleaned using the GenElute kit (Sigma-Aldrich) and sequenced by Sanger sequencing. The sequence was used to determine the position and orientation of the Ds insertion in the *C. albicans* genome.

## Growth assays

For hemoglobin utilization, overnight cultures grown in YPD were diluted in the morning into a series of two-fold dilutions of hemoglobin in YPD + ferrozine or BPS. Cells were inoculated in flat-bottomed 96-well plates at $OD_{600}$ = 0.00001, 150 µl per well. Plates were incubated at 30˚C on an orbital shaker at 60 rpm and growth was measured by optical density ($OD_{600}$) after 2 and 3 days with an ELISA reader. Cells were resuspended with a multi-pipettor before each reading. Each culture was done in triplicate. For MPP sensitivity, cells were inoculated in flat-bottomed 96-well plates in YPD + 1 mM ferrozine at $OD_{600}$ = 0.0001, 150 µl per well. Plates were incubated at 30˚C on an orbital shaker at 60 rpm and growth was measured by optical density ($OD_{600}$) after 2 days with an ELISA reader.

## Heme sensor binding assays

The protocol was as described [14]. Briefly, each strain to be tested was transformed with the sensor plasmids KB2636 (WT HS1), KB2669 (M7A), as well as a vector plasmid. Overnight cultures were diluted to O.D. $_{600}$ = 0.2 and grown for 4h in the indicated hemin concentrations. For fluorescence measurements, cells were pelleted, washed once with PBS, resuspended to O.D. $_{600}$ = 5, and 0.2 ml were placed in duplicate in a black 96 well flat bottom plates (Nunc Fluorotrac). Fluorescence was measured with a Tecan infinite 200 Pro reader, with eGFP: ex. 480 nm (9 nm bandwidth), em. 520 nm (20 nm bandwidth), and mKATE2: ex. 588 nm (9 nm bandwidth), em. 620 nm (20 nm bandwidth). Each strain was represented by three independent cultures, and each culture was measured twice (technical duplicate), the vector-only culture reading was substracted from all other readings, and the ratio of eGFP to mKATE2 was calculated.

## Glucose chase and Western blotting

For glucose chase of Hap1-6xMyc, cells were grown overnight in YEP + 2% raffinose at 30˚C, and diluted in the morning to $OD_{600}$ = 0.5, grown for 3 hours in YEP + 2% maltose for induction, with 20 μM hemin or without hemin, then 2% glucose was added and 1 ml aliquots were taken at 0', 30', 60' and 90'. Samples were then prepared as described [58]. Briefly, aliquots were treated with 0.25 N NaOH, 1% 2-mercaptoethanol, TCA-precipitated, washed in acetone and resuspended in protein loading buffer, then separated by SDS-PAGE and Western blotting. The blots were reacted with the 9E10 anti-Myc monoclonal antibody, and with monoclonal anti-tubulin (DM1A, Sigma) as internal control, and the bands were visualized by ECL and quantitated using an a ImageQuant LAS4010 apparatus. For detection of Rbt5, a previously described rabbit polyclonal antiserum [16] was used, and C4 anti-actin monoclonal antibody (MP Biomedicals) as internal control, and quantitated as above.

## Microscopy

Cells were imaged with a Zeiss Axioskop Imager epifluorescence microscope equipped with DIC optics, using a 100X objective. For visualizing Hap1-GFP, the GFP filter set was used with an exposure time of 6 seconds.

Nuclear stain: Hoechst 33342 was added to an exponentially growing culture to 10 μg/ml from a 1 mg/ml stock in water and the cells were incubated a further 30 min at 30˚C prior to microscopy visualization. For visualizing Hoechst 33342 by epifluorescence microscopy, the DAPI filter set was used with an exposure time of 1 second.

## RNA extraction and analysis

For RT-PCR, for each sample 10 ml of culture was spun down and the pellet was frozen in liquid nitrogen and stored at -80˚C. The pellet was re-suspended in 400 μl lysis buffer (10 mM Tris-HCl pH 7.5, 10 mM EDTA, 0.5% SDS), and 400 μl acidic phenol, vortexed and incubated at 65˚C for 1 hour with occasional vortexing ("hot phenol" method [59]). The tubes were then placed on ice for 10 min and then centrifuged for 10 min at 4˚C, max speed. The upper layer was removed to a new tube, re-extracted with equal volume of phenol-chloroform 1:5 and vortexed. The tubes were centrifuged again for 10 min and the upper layer was removed into new tube, re-extracted with equal volume of chloroform and vortexed. The tubes were centrifuged again for 10 min and the upper layer was removed into new tube, for ethanol precipitation. Next, samples were treated with DNase to remove contaminating DNA from RNA using TURBO DNA-free Kit according to the manufacturer's instructions, followed by cDNA

synthesis from 1000 ng of RNA, according to the Applied Biosystems High-Capacity RNA-to-cDNA Kit in total 20 μl reaction. RT-PCR (primers 15–28) was performed using QuantStudio 3 Real-Time PCR System equipment with Fast SYBR Green Master Mix in a final volume 10 μl. Expression in each sample was normalized to *ACT1* expression.

For RNA-SEQ, mutant and reintegrant cells were grown overnight in YPD, diluted in the morning in YPD to $OD_{600} = 0.5$. After 2.5 h at 30˚, the cultures were split in two and 50 μM hemin was added from a 2 mM stock, and samples were taken after 30 min. RNA was extracted using the "hot phenol" method [59]. Library construction and analysis was performed at the BCF Technion Genomics Center with NEBNext Ultra II Directional RNA Library Prep Kit for Illumina, cat no. E7760, using 800ng total RNA as a starting material. mRNAs pull-down was performed using the Magnetic Isolation Module (NEB, cat no. E7490). After construction, the concentration of each library was measured using Qubit (Invitrogen) and the size was determined using the TapeStation 4200 with the High Sensitivity D1000 kit (cat no. 5067–5584). All libraries were mixed into a single tube with equal molarity. The RNAseq data was generated on Illumina NextSeq2000, using P2 100 cycles (Read1-100; Index1-8; Index2-8) (Illumina, cat no. 20046811). Quality control was assessed using Fastqc (v0.11.8), reads were trimmed for adapters, low quality 3'and minimum length of 20 using CUTADAPT (v1.10). 100 bp single reads were aligned to a *Candida albicans* reference genome (GCF_000182965.3_ASM18296v3_genomic.fna from NCBI) using Tophat (v2.1.0, uses Bowtie2 version 2.2.6). Between $26 \times 10^6$ and $31 \times 10^6$ uniquely mapped reads (~96% of total reads) were obtained per sample. The number of reads per gene was counted using HTSeq-count (v2.2.0). A statistical analysis was preformed using DESeq2 R package (v1.28.0) [60]. The number of reads per gene was extracted into merged_counts.csv and normalized_counts.csv files for raw counts and normalized counts, respectively. The similarity between samples was evaluated within DESeq2 package using correlation matrix, shown in two plots- heatmap and principal component analysis (PCA). Raw data are available at NCBI's GEO database under accession number GSE202577.

GO analysis on "Biological Process" was carried out using PANTHER [61] (http://geneontology.org/), but using the list of 5921 protein-coding genes detected in our experiment as reference, rather than the total list of genes. Intersects between lists were obtained and Venn diagrams were drawn with Venny 2.1 (https://bioinfogp.cnb.csic.es/tools/venny/index.html).

## Sequence comparison

The *S. cerevisiae* Hap1 and *C. albicans* Zcf20/Hap1 sequences were used to identify homologs in Saccharomycetales genomes using the Candida Genome Database (www.candidagenome.org) and Saccharomyces Genome Database (www.yeastgenome.org). 46 closest matches in 38 genomes (S1 Data) were aligned using the MAFFT G-INS-i algorithm [63], followed by tree building using the NJ method with 100 X resampling. Similar trees were obtained with other algorithms (UPGMA) and aligning methods (NCBI COBALT).

## Supporting information

**S1 Fig. Alignment of the *S. cerevisiae* Hap1 sequence with the *C. albicans* Hap1/Zcf20 sequence.** The *S. cerevisiae* Hap1 (top) was aligned with *C. albicans* Hap1/Zcf20 (bottom) using the MAFFT G-INS-i algorithm [63] with homologs. The DNA-binding and dimerization domains, the repressing modules (RPM1-3), and the heme-responsive motifs (HRM1-7) [33,35,64], of Hap1 are indicated above the sequence. The HRM consensus sequences R/KCPV/I in both proteins are highlighted.
(JPG)

**S2 Fig. Proximity tree of sequences related to *S. cerevisiae* Hap1 and *C. albicans* Hap1/Zcf20.** The Hap1 protein sequences of *S. cerevisiae* and *C. albicans* were used to identify the 44 most related sequences in 38 Saccharomycetales genomes and the sequences were aligned, followed by tree building. The red rectangle indicates sequences containing the HRM motif R/KCPV/I and the purple rectangle indicates sequences lacking it.
(JPG)

**S3 Fig. Alignment of Heme Regulatory Motifs in Hap1 homologs.** 38 Hap1 protein sequences identified by homology to the *S. cerevisiae* and *C. albicans* Hap1 sequences were aligned, and regions with conserved K/RCPV/I motifs were visualized (indicated with red rectangles). The *S. cerevisiae* Hap1 HRM1-7 motifs are indicated at the bottom. Residues conserved in more than half the sequences are highlighted. The species marked in green font belong to the CUG-Ser1 clade, the species marked in blue font belong to the Saccharomycetaceae and those in purple font, to the Pichiaceae.
(JPG)

**S4 Fig. The *hap1*$^{-/-}$ mutant is defective in growth on hemoglobin in RPMI medium supplemented with transferrin.** Two independent *hap1*$^{-/-}$ mutant clones from the Homann collection [36], together with wild-type strain SN148, were inoculated in desferrated RPMI medium supplemented with transferrin and with increasing amounts of hemoglobin, as indicated. The wild-type and mutant strains were inoculated in triplicate in 96 well plates, and incubated for 2 days at 30˚C. The graph indicates the average density for each triplicate, and the error bars indicate the standard deviations.
(JPG)

**S5 Fig. The *hap1*$^{-/-}$ mutant grows faster in the presence of GaPPIX.** The wild-type strain SN148 and two independent *hap1*$^{-/-}$ mutant strains (KC1406) were inoculated at $OD_{600}$ = 0.0005 in YPD supplemented with 1 mM ferrozine and 150 μM GaPPIX, and incubated at 30˚C in 50 ml Falcon tubes with vigorous shaking. The densities were measured at the indicated times after inoculation.
(JPG)

**S1 Table. Pairwise comparisons of expression of genes in *HAP1* and *hap1*$^{-/-}$, with or without added hemin.**
(XLSX)

**S2 Table. Genes significantly up in the four pairwise comparisons.**
(XLSX)

**S3 Table. Genes significantly down in the four pairwise comparisons.**
(XLSX)

**S4 Table. GO enrichment analysis of genes at least 2x up in the four pairwise comparisons.**
(XLSX)

**S5 Table. GO enrichment analysis of genes at least 2x down in the four pairwise comparisons.**
(XLSX)

**S6 Table. Patmatch analysis of the Hap1 binding motif in the *C. albicans* intergenic regions.**
(XLSX)

**S7 Table. List of primers.**
(DOCX)

**S1 Data. Sequences of Hap1 homologs.**
(TXT)

**S2 Data. Numeric data of all graphs.**
(XLSX)

## Acknowledgments

We thank Liat Linde and Maor Hatoel (Biomedical Core Facility Genomics Center, Technion—B. Rappaport Faculty of Medicine) for carrying out RNA-seq analysis, and Sara Selig for critical reading of the manuscript.

## Author Contributions

**Conceptualization:** Daniel Kornitzer.

**Data curation:** Mariel Pinsky, Daniel Kornitzer.

**Formal analysis:** Ziva Weissman, Mariel Pinsky, Daniel Kornitzer.

**Funding acquisition:** Daniel Kornitzer.

**Investigation:** Natalie Andrawes, Ziva Weissman, Mariel Pinsky, Shilat Moshe.

**Project administration:** Ziva Weissman, Daniel Kornitzer.

**Resources:** Judith Berman.

**Supervision:** Ziva Weissman, Daniel Kornitzer.

**Visualization:** Natalie Andrawes.

**Writing – original draft:** Natalie Andrawes, Daniel Kornitzer.

**Writing – review & editing:** Judith Berman, Daniel Kornitzer.

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
