## [Decision Letter · Decision Letter 0]

18 Jul 2022

Dear Dr Kornitzer,

Thank you very much for submitting your Research Article entitled 'Regulation of heme utilization and homeostasis in Candida albicans' to PLOS Genetics.

The manuscript was fully evaluated at the editorial level and by independent peer reviewers. The reviewers appreciated the attention to an important topic but they identified some concerns that we ask you address in a revised manuscript. Most concerns van be addresses by clarifying the text.

Firstly, I would like you to reconsider naming the gene "Huf1". You have clearly shown that Huf1 is an ortholog of Hap1 in S. cereviaise, and the field can get very confused when different gene names are used in different species. Your manuscript is likely to appeal to a larger audience if it is clear that you are describing species-specific functions of Hap1.

Secondly, major new additional experiments suggested by the reviewers (such as confirming binding of Huf1, or overexpressing HUF1) are not necessary providing that the text is clear. You could consider assaying catalase as suggested by Reviewer 2. You should also clarify heme/hemin concentrations, confirm that the the number of replicates used is suitable, and clarify fold-changes discussed as suggested by the reviewers. Most other concerns can be addressed by changing some text, moving some material from supplementary to the main text, and for example referring to supplementary tables in the legend of Fig.6.

We therefore ask you to modify the manuscript according to the review recommendations. 

[LINK]

Yours sincerely,

Geraldine Butler

Section Editor: Prokaryotic Genetics

PLOS Genetics

Gregory Copenhaver

Editor-in-Chief

PLOS Genetics

Reviewer's Responses to Questions

**Comments to the Authors:**

Reviewer #1: (1) Line 160. Although the presence of the consensus ScHap1 in the promoter regions of some genes, there is no direct evidence that Huf1 directly binds to these promoters to promote transcription. This uncertainty should be mentioned i the text.

(2) Something seems missing in the model shown in Figure 8. The coactivation of heme uptake and degradation is apparently not economical for the cell unless the main purpose of degradation is releasing iron for storage in addition to preventing heme toxicity. Figure 3 shows that the HMX1 expression level is at least 50 times higher than FRP2 in the absence of hemin. How will these data fit into the model?

Reviewer #2: This manuscript describes a genetic dissection of the Huf1 transcription factor of Candida albicans. At the beginning, the authors used a stable haploid C. albicans strain that was mutagenized with the Ac/Ds transposon to isolate GaPPIX-resistant mutants. Based on their reduced ability to grow on hemoglobin as the sole iron source, several mutants were characterized for their transposon insertion site. Among them, the authors identified that inactivation of the HUF1 gene in a standard diploid C. albicans strain recapitulates the earlier phenotype. Using fluorescent heme sensors, the absence of Huf1 results in higher cytoplasmic heme concentrations. The authors establish that HUF1 is transcriptionally induced in the presence of hemin. They show that Huf1 is a nuclear protein that is stabilized in the presence of hemin. The authors used RNA-Seq to investigate the transcriptional response under hemin-replete conditions and identified several genes that exhibit Huf1-dependent changes on a genome-wide scale. Overall, the findings will certainly be of interest to the field of microbial heme homeostasis and fungal pathogens.

Comments.

1. A schematic illustration of C. albicans Huf1 with Saccharomyces cerevisiae Hap1 would be important to add as a figure in the manuscript. In this way, location of conserved domains such as DNA-binding and heme-binding domains of these transcription factors could be shown. Percentages of identity and similarity between these two proteins would also be very informative.

2. Is there any experimental evidence that Huf1 or a portion of Huf1 directly binds hemin? Is there absorbance spectroscopy that has been performed to detect changes at the Soret peak?

3. According to the proposed model (Fig. 8), Huf1 activates genes encoding proteins involved in heme acquisition and degradation. Do the authors have performed ChIP analysis to test whether Huf1 exhibits differential binding as a function of the presence or absence of hemin and/or iron availability?

4. The authors conclude that the DNA binding motif recognized by Huf1 is to some extent similar to that of Hap1, which binds to a DNA sequence CGGNNNTANCGG. Do the authors have experimental data to support this conclusion? Do the authors have used promoter-lacZ fusions to map cis-acting elements necessary for hemin- and Huf1-dependent activation of gene expression?

5. Does overexpression of Huf1 enhance hemin or heme analog acquisition? Does overexpression of Huf1 lower cytoplasmic heme levels through measurement using cytoplasmic fluorescent heme sensors?

6. In Fig. 4, what would happen if wild-type HUF1 is returned in huf1-/- cells? Does it restore cytoplasmic heme to wild-type levels? In huf1-/- cells, cytoplasmic heme levels are higher, do huf1-/- cells exhibit normal catalase activity? This would be very interesting to assess since in Fig. 6B, the CAT1 gene (encoding catalase) appears to be a Huf1 target gene.

7. In the text that describes results in Fig. 4, especially in lines 187 to 212, fold values are not indicated and are not described with enough detail. These data should be commented with more detail since without a clear description, it is very difficult to understand and therefore appreciate the results.

8. In Fig S3 and Fig 5, why are three different concentrations of hemin that were used? Experiments in Fig S3 were performed using 10 uM hemin, whereas experiments in Fig 5 A and C were performed using 50 uM and 20 uM hemin, respectively. How could you compare the data if different hemin concentrations have been used?

9. In Fig 5B, it is unclear whether there is significant increase of the nuclear Huf1-GFP signal in the presence of GaPPIX and MnPPIX as compared with untreated cells? In the case of GaPPIX, it is intriguing that the signal is so low since the original screen was performed to identify GaPPIX-resistant clones. I would recommend to specify that no signal is seen in the presence of MnPPIX.

10. In Fig. 5C, only three time points have been taken after glucose addition, with only a single time point (90 min) that indicates a significant difference. It would be very informative to take additional measurements as a function of time (longer time points like Fig 7B in which 2, 4, and 6 h time points have been taken), especially for assessment of the protein levels of Huf1 in the absence or presence of hemin.

11. Fig. 6B would beneficiate to be shown as regular Table(s) in the manuscript to list and describe all target genes. The current Venn diagram with color codes is very difficult and confusing to understand for the reader.

Reviewer #3: This is an excellent manuscript demonstrating for the first time, a heme regulated transcription factor in the opportunistic fungal pathogen Candida albicans. While a great deal is understood regarding the complex control of C. albicans Fe homeostasis through the SFU1-SEF1-HAP43 regulatory network, this is the first demonstration of a separate transcription factor dedicated to heme uptake. Most interesting, the ortholog of this factor in the non-pathogenic S. cerevisiae does not participate in heme uptake as heme is not an abundant source of Fe for this organism. The repurposing of this transcription (now called HUF1 in C. albicans) is a clever adaptation of the pathogenic yeast to existence in an animal host. The paper is rigorous, comprehensive and written nicely for a diverse audience. I only have a few relatively minor suggestions for improvement.

1. Statistics are needed for Fig. 4A and Fig. 4B. The authors discuss differences in the huf1 mutant versus WT strain that are not obvious in the graphs provided without the appropriate analysis to show these changes are statistically significant. In the same figure, the authors say the results are the averages of three cultures. Does this mean the experiment was done only once with three cultures?

2. In the legend to Fig. 4, it would be helpful to define HS1 (heme sensor), M7A (lower heme affinity probe) and DM (non heme binding control).

3. Lines 221 – 224 and line 369 mention “regular media”. What does "regular" mean, specifically in terms of available iron and heme sources?

4. Is it possible to insert the data of Fig. S3 into main Fig. 5. The title of the section is “Huf1 is nuclear localized…” but one has to download supplemental material to see the data that support this important claim.

**Have all data underlying the figures and results presented in the manuscript been provided?**

Reviewer #1: Yes

Reviewer #2: Yes

Reviewer #3: Yes

PLOS authors have the option to publish the peer review history of their article (what does this mean?). If published, this will include your full peer review and any attached files.

Reviewer #1: No

Reviewer #2: No

Reviewer #3: No

---

## [Editor Report · Decision Letter 1]

22 Aug 2022

Dear Dr Kornitzer,

Thank you for carefully addressing the reviewers' comments. We are pleased to inform you that your manuscript entitled "Regulation of heme utilization and homeostasis in Candida albicans" has been editorially accepted for publication in PLOS Genetics. Congratulations!

When making formatting changes, I suggest that you replace the word "homology"  with "sequence similarity" in the third paragraph of the results.

Yours sincerely,

Geraldine Butler

Section Editor

PLOS Genetics

Gregory Copenhaver

Editor-in-Chief

PLOS Genetics

Comments from the reviewers (if applicable):

**Data Deposition**

http://datadryad.org/submit?journalID=pgenetics&manu=PGENETICS-D-22-00736R1

**Press Queries**

---

## [Editor Report · Acceptance letter]

5 Sep 2022

PGENETICS-D-22-00736R1 

Regulation of heme utilization and homeostasis in Candida albicans 

Dear Dr Kornitzer, 

We are pleased to inform you that your manuscript entitled "Regulation of heme utilization and homeostasis in Candida albicans" has been formally accepted for publication in PLOS Genetics! Your manuscript is now with our production department and you will be notified of the publication date in due course.

With kind regards,

Anita Estes

PLOS Genetics

On behalf of:
